# Formulation and Evaluation of a Clove Oil-Encapsulated Nanofiber Formulation for Effective Wound-Healing

**DOI:** 10.3390/molecules26092491

**Published:** 2021-04-24

**Authors:** Misbah Hameed, Akhtar Rasul, Muhammad Khurram Waqas, Malik Saadullah, Nosheen Aslam, Ghulam Abbas, Sumera Latif, Hafsa Afzal, Sana Inam, Pervaiz Akhtar Shah

**Affiliations:** 1Department of Pharmaceutics, Government College University Faisalabad, Faisalabad 38000, Pakistan; misbahmajid1@gmail.com (M.H.); ghulamabbas@gcuf.edu.pk (G.A.); drsanainam@gmail.com (S.I.); 2Institute of Pharmaceutical Sciences, University of Veterinary and Animal Sciences, Lahore 54000, Pakistan; mkhurramwaqs@gmail.com; 3Department of Pharmaceutical Chemistry, Government College University Faisalabad, Faisalabad 38000, Pakistan; maliksaadullah@gcuf.edu.pk; 4Department of Biochemistry, Government College University Faisalabad, Faisalabad 38000, Pakistan; afzal_nosheen@yahoo.com; 5Institute of Pharmacy, Faculty of Pharmaceutical and Allied Health sciences, Lahore College for Women University, Lahore 54000, Pakistan; sumera_latif@hotmail.com (S.L.); hafsa382@gmail.com (H.A.); 6College of Pharmacy, University of the Punjab, Lahore 54000, Pakistan; pashah6512@yahoo.com

**Keywords:** nanofibers (NFs), wound-healing, antibacterial, clove bud

## Abstract

Wound-healing is complicated process that is affected by many factors, especially bacterial infiltration at the site and not only the need for the regeneration of damaged tissues but also the requirement for antibacterial, anti-inflammatory, and analgesic activity at the injured site. The objective of the present study was to develop and evaluate the natural essential oil-containing nanofiber (NF) mat with enhanced antibacterial activity, regenerative, non-cytotoxic, and wound-healing potential. Clove essential oil (CEO) encapsulated in chitosan and poly-ethylene oxide (PEO) polymers to form NFs and their morphology was analyzed using scanning electron microscopy (SEM) that confirmed the finest NFs prepared with a diameter of 154 ± 35 nm. The successful incorporation of CEO was characterized by Fourier transform infra-red spectroscopy (FTIR) and X-ray diffractometry (XRD). The 87.6 ± 13.1% encapsulation efficiency and 8.9 ± 0.98% loading of CEO was observed. A total of 79% release of CEO was observed in acidic pH 5.5 with 117% high degree of swelling. The prepared NF mat showed good antibacterial activity against Staphylococcus aureus and Escherichia coli and non-cytotoxic behavior against human fibroblast cell lines and showed good wound-healing potential.

## 1. Introduction

Growth and infiltration of an open wound by microorganisms is one of the major factors that delays the wound-healing process and may result in skin deformation. Thus, apart from healing the wound, another critical issue to be addressed is to prevent the wound from microbial contamination [1,2]. For this, there is a need for a dressing that provides antibacterial protection at the site of application along with a suitably moist environment to hasten the healing process. Recently phototherapeutics have been explored for their multiple therapeutic benefits, including their healing and antimicrobial capacity [3].

Clove bud (Syzygium aromaticum) of the Myrtaceae family is considered to be one of the most effective and influential antimicrobial natural herbs. It is indigenous to Indonesia, Sri Lanka, and India. Studies have shown that essential oil of clove has good antibacterial, antifungal, antioxidant, analgesic, anesthetic, and insecticidal activity [4,5]. Eugenol, eugenol acetate, and β-caryophyllene, being major components, are responsible for its various therapeutic effects [6]. The FDA has categorized clove oil as generally recognized as safe (GRAS); however, it is lethal when taken via an oral dose of 3.75 g/kg body weight. Clove oil becomes yellowish and is chemically unstable in air; moisture, light and temperature during storage cause highly volatile losses its activity [7]. Owing to its antibacterial, antioxidant, anti-inflammatory, and analgesic properties, clove oil is one of the ideal essential oils for the healing of wounds. Clove essential oil fabricated as NFs with various polymers show strong antibacterial properties, especially against Staphylococcus aureus, Escherichia coli, Pseudomonas fluorescens, and Bacillus subtilis [8,9,10]. The wound-healing effect of clove oil in nanoemulsion form was studied, and it was confirmed that the oil in nanoemulsion has a marked wound-healing capacity and reduced the incidence of inflammatory cells at the site of a wound when compared to pure clove oil and enhanced cell viability [11,12]. Poly(ε-Caprolactone)/Gelatin NFs loaded with clove oil showed good antibacterial as well as marked healing properties [10]. Clove oil-based nanofibrous mats showed good antifungal properties when used for candida-associated denture stomatitis prevention and treatment [13].

Chitosan is a natural polymer that is biodegradable, biocompatible, and has good antibacterial properties. Chitosan and its derivatives have been studied for their antibacterial, antioxidant, analgesic, and antitumor properties. The developed formulations of chitosan have been used for drug and gene delivery, tissue and scaffold engineering, and wound-healing [14,15]. Chitosan has been given regulatory approval in the USA for its use in wound-healing bandages and other healing formulation. However, the major drawback is its poor spinnability. Owing to its structural and poor mechanical properties, it cannot be used alone in electrospinning [16]. When dissolved in acids such as acetic acid, CS becomes polyelectrolyte and resisted the formation of NFs because of repulsion caused between the ionic groups in the backbone, and the degree of deacetylation also effects the polyelectrolyte nature of CS. The higher the degree of deacetylation, the higher the polyelectrolyte nature of the CS [17,18]. Although some researchers have reported chitosan composite fibers, usually they are prepared when combined with some hydrophilic synthetic copolymer polyvinyl alcohol and polyethyleneoxide (PEO) [19,20]. The aim of this research was to prepare clove oil-based NF dressing using a blend of chitosan and poly-ethylene oxide and observe their synergistic effect as antibacterial agents and effective wound-healing candidates.

## 2. Results and Discussion

### 2.1. Constituents of Clove Essential Oil

GC-MS was performed to determine the composition of the CEO using internal standard method. The results are given in Table 1, showing the retention time (minutes) and percentage composition of various constituents. CEO is a mixture of phenylpropanoids, monoterpenoids, and sesquiterpenoids, with a small percentage of alcohols, aldehydes, and ketones. In total, 33 different constituents were identified using GC-MS analysis. The major constituents determined were eugenol, eugenol acetate, β-caryophyllene, and β-phellandrene.

### 2.2. Morphology and Fiber Diameter

The morphology of the fibers is given in Figure 1. In the preparation of CEO-loaded NFs, PEO was used in combination with CS as CS alone is not spinnable owing to its structural entanglements. Amounts of 0.5% and 1% CEO were added to the spinning solution. Figure 1A is the fiber formed using 0.5% CEO, and 1B resulted from 1% of CEO. The fiber diameter was measured using ImageJ software, which showed that by increasing the percentage of CEO, the fiber formation was not affected; however, the thickness of fibers increased from 154±35nm resulting from 0.5% CEO to 189 ± 43 nm resulting from 1% CEO. The increase in the thickness of fiber as shown in Figure 2 may be due to a decrease in electrical conductivity of the solution and increase in solution viscosity because of some interactions between the CEO and PEO [10,21]. Similar results were also reported in a study where the diameter of NFs increased by increasing the percentage of loaded clove essential oil from 1.5–6% (*w*/*v*) [10]. Fish oil was encapsulated in polyvinyl alcohol NFs and reported that by increasing the percentage of loaded oil, the thickness of fibers increased [22]. Candeia essential oil was encapsulated in polylactic acid and it was found that increased average diameter of fibers formed by increasing the concentration of candeia oil [23]. However, another study reported that during the fabrication of peppermint oil nanocomposites using polycaprolactone, the average diameter of the fibers decreased by using a high percentage of the oil [10]. It is observed from Table 2 that an increase in the concentration of CS results in an increase in viscosity and decrease in conductivity; this change in viscosity and conductivity also affected fiber formation. Only a 50:50 ratio of CS and PEO resulted in smooth, beadless fibers. As the concentration of CS increases, the solution becomes more viscous and difficult to spin, resulting in beaded structures.

### 2.3. FTIR Analysis

FTIR analysis of CS, PEO, Blank NFs, and CEO-loaded NFs showing the chemical characterization and functional groups are shown in Figure 3. Changes in the CEO NF spectrum was observed at 2904 cm^−1^ and 1538 cm^−1^ indicating the clove oil incorporation in the NFs. The CS spectra showed a broad peak at 3248 cm^−1^ due to overlapping and stretching of NH2 and –OH. Transmittance peak appeared at 2883 cm^−1^ owing to symmetric stretching of the CH group of the pyranose structure. Deep peaks appeared at 1648 cm^−1^ due to deformational vibrations of C–O group of primary amines. A peak at 1530 cm^−1^ is due to the bending of NH2 groups. A deep peak at 1410 cm^−1^ corresponded to the bending vibrations of OH and CH [24].

PEO spectrum analysis showed a sharp peak at 2875 cm^−1^ and 1344 cm^−1^ that corresponded to the symmetric CH and CH2 stretching and bending. CN stretching associated with pyranose ring and COC vibration in PEO structure is confirmed by the peaks at 1347 cm^−1^ and 1097 cm^−1^, respectively. Other peaks at 1102 cm^−1^ and 962 cm^−1^ are attributed to the asymmetric stretching of the C–O group [25,26]. FTIR spectra of clove oil showed the peak at 3022 due to O–H stretching. Peaks at 1621 cm^−1^ and 1516 cm^−1^ corresponded to the main peaks due to eugenol. Peaks at 1443 cm^−1^ were due to C–C stretching vibrations of the phenyl ring while the one at 1178 cm^−1^ was due to C–O bending. The spectrum of clove oil-loaded NFs showed characteristic peaks at 3015 cm^−1^, 1505 cm^−1^, and 1164 cm^−1^ showing the successful incorporation of the oil in the nanoweb. 

### 2.4. XRD Analysis

XRD analysis is shown in Figure 4. CS is amorphous in nature, thus did not show any sharp spike, but has a broad peak at 2θ of 18–220 which is because of its degree of crystallinity due to its amorphous shape [27,28]. PEO shows some sharp spikes at 19.12, 23.25, 26.17, and 26.87, which is attribute to its crystalline nature. In the case of CEO, there is one broad peak at 11.45–13.35 that does not show any sharp spike in any other position, it being a viscous liquid in structure. The distinguishing peaks corresponding to the oil and PEO were also observed in the pattern produced by the physical mixture. This suggested that the crystalline components of PEO maintained their crystalline nature in the physical mixture. The XRD pattern of CEO NFs suggests that the diffraction spectrum with the characteristic peak of at 2θ of 19° and 23° confirms the presence of CEO within NFs, as well as the incorporation of PEO as crystalline or semi-crystalline form. The XRD analysis was done to identify the phase transformations and probable interactions due to the infusion of CEO. XRD patterns, as shown in the figure, of the NFs with and without CEO showed diffraction peaks at the same 2θ, and this means there is no phase transformation and reaction because of inclusion with CEO. A similar result was reported during the encapsulation of tea-tree oil. It was reported earlier that concentration of the polymer in the solution largely determines the crystallinity of the NFs formed [23].

### 2.5. Percentage Oil Content, Drug-Loading Efficiency (DLE%), and Percentage Yield

CEO CS-PEO NFs showed 87.6% drug content. The drug-loading may be attributed to the hydrophobic nature of the oil and may be because of the little evaporation of the oil during the process of electrospinning. CEO loading was 8.9% while percentage yield was 79% as shown in Table 3. It can be seen from Table 3 that the NFs showed high percentage drug content. Drug content loaded in the NFs depends on various factors such as the nature of the drug and polymer (hydrophilic/hydrophobic), compatibility between the drug loaded and the polymer, and upon the method used for the loading of the drug [29]. The high drug content loaded may be due to some ionic interactions between the drug and the polymer and polymer system used for entrapment of the drug. One of the reasons for the encapsulation of essential oils is to improve the encapsulation efficiency of volatile components. Eugenol, the major component of clove oil, shows high volatility and poor water solubility [30,31,32,33]. High loading of clove oil was earlier reported by Tonglairoum et al., in 2016 [13]. A total of 73% encapsulation of CEO in PCL NFs was reported by Unalan et al., in 2019 [10]. In another study, CEO chitosan NPs were loaded in gelatin NFs with varied encapsulation efficiency 21.1 ± 0.4% to 39.6 ± 0.8% [34].

### 2.6. Release of CEO from NFs 

The rate of release of loaded oil from NFs is shown in Figure 5. The release rate was measured at two pH conditions. The release profile was biphasic, showing faster release in the initial 12 hours where approximately 60% of the loaded drug was released, followed by slow and continuous release for the subsequent 36 hours. Initial high release might be caused by hydrophilic behavior of PEO and followed by sustained release due to the presence of CS. As can be seen from the figure, the release rate of CEO is faster in acidic pH (5.5) as compared to physiologic pH (7.4) which might be attributed to a high affinity of CS to dissolve in acidic conditions. Moreover, high release in acidic conditions is also useful in topical formulations and shows the potential to be used topically [35,36]. It was observed that during the initial 12 hours, the release was faster at both pHs; the release profiles were not significantly different (*p* > 0.05) from each other over this period, but showed significant difference (*p* < 0.05) after 12 hours. The same results were reported for ciprofloxacin-encapsulated NFs developed for topical burn-wound-healing [37].

### 2.7. Water Absorption Capacity

Water absorption capacity is important for the release of the drug at the wound site. Also, it is responsible for the absorption of excessive exudate oozing from the wound. Both these factors enhance the effectiveness of wound management. Water absorption capacity can be assessed by the determining degree of swelling. It was observed that the degree of swelling of samples varied when calculated before and after the addition of drug in NFs. PEO-CS NFs were taken as a control. The degree of swelling found in the blank CS-PEO polymeric composite NFs was 120 ± 17%. However, for CEO-loaded CS-PEO NF water absorption capacity was less i.e., 117 ± 4%, than that of blank NFs, as shown in Figure 6A. The addition of oil slightly decreases the water absorption capacity of NFs, and this may be attributed to the hydrophobic nature of the oil added. The presence of the oil in NFs may cause a hindrance in the interaction of the polymer with the water and thus reducing the overall water absorption from the solution [3,38]. The overall absorption of water by the CS-PEO NFs can also be attributed to the presence of high concentration of PEO, which tends to form a gel by absorption of water at pH 7.4, while, on the other hand, CS shows less solubility at pH 7.4. However, an increase in the wettability was reported by increasing the concentration of CEO [10]. Cinnamon EO-encapsulated chitosan NPs incorporated in PLA nanocomposite resulted in an increase of the hydrophilic behavior of the NFs [39]. PCL aloe vera electrospun NFs showed a decrease in hydrophilic properties [40].

### 2.8. Antibacterial Activity

The qualitative antibacterial activity of blank and CEO-loaded NFs was measured by calculating zones of inhibition against selected test strains. The results showed that both blank CEO-loaded NFs had an effective antibacterial activity. The blank NFs showed a zone of inhibition of 24.3 ± 3.40%, 23.55 ± 5.27%, and 23.1 ± 4.6%, while CEO NFs showed 36.6 ± 2.4%, 36.2 ± 3.5%, and 33.6 ± 5.4% against *Escherichia coli*, *Staphylococcus aureus*, and *Pseudomonas aeruginosa* respectively, as shown in Figure 7. The higher activity of NFs against *E.coli* could be attributed to the structural difference of cell wall. The antibacterial activity of clove has also been evaluated earlier [8]. 

### 2.9. Cytotoxicity Evaluation

Cytotoxicity evaluation of the prepared NFs was done using fibroblast cell lines after 48 h of incubation. Cell viability was measured by MTT assay, which showed no significant difference between cytotoxicity of blank NFs, CEO NFs, and control (*p* > 0.05), suggesting that the CEO, along with the polymers used, is safe at the selected concentration with no cytotoxicity against fibroblasts cell lines as shown in Figure 6B. Cytotoxicity of CEO PCL NFs against NHDF cells was evaluated and reported to be non-cytotoxic in nature [10]. In another study conducted using lavender oil loaded in alginate, NFs reported 91% cell viability when tested on HFF-1 cell lines [41]. Peppermint and chamomile oil-loaded gelatin NFs were tested against NIH-3T3 fibroblast cell lines and the results confirmed no change in cell viability as compared to control. Another study evaluated the cytotoxicity of CEO with increasing concentration ranging from 0.0001 to 10 mg/ml and observed that CEO has a cytotoxic effect at high concentrations, however at lower concentrations it was safe at 0.0001 to 1 mg/ml for 2 h and at 0.0001 to 0.1 mg/ml for 24 h [13].

### 2.10. Wound-Healing Potential 

The representative images of wounds on day 1, day 5, and day 10 are shown in Figure 8. 

The wound size decreases slowly over a period of time. Wound-healing was slowest in untreated group (A), showing the maximum diameter throughout the study period with wound closure 17.56% and 50.44% on day 5 and 10, respectively, while Group D, treated with commercial product, showed maximum healing of 49.1% and 95.3%, as shown in Figure 9, which showed highly significant (*p* < 0.001) results when compared to untreated groups, whereas Group C, treated with the CEO NFs, also showed better healing percentage (45.41 and 90.13) on the 5th and 10th day of treatment, which is more significantly different (*p* < 0.01) from Group A, but less than that of the commercial product, as shown in Table 4. On the other hand, Group B, treated with blank NF formulation, showed some healing properties, which can be attributed to the presence of CS in the NFs confirming its healing capacity resulting from its antibacterial and antioxidant properties. Wound-healing potential of clove oil was also observed using in vitro scratch assay and confirmed the healing potential [10].

## 3. Materials and Methods

### 3.1. Materials

Chitosan was sourced from Sigma–Aldrich Gmbh Darmstadt, Germany and acetic acid was purchased from DAEJUNG, Gyeonggi-do, South Korea. PEO (Mw 600KD) was purchased from Polysciences, Inc., Niles, IL, USA. Clove essential oil (CEO) was purchased from Mueller Hinton broth Sigma–Aldrich Gmbh, Darmstadt, Germany. All chemicals were used without any further purification.

### 3.2. Methodology

Components of the CEO were determined by gas chromatography mass spectrometry (GC-MS). GC equipped with HP-5MS capillary column (30 m × 0.25 mm i.d., film thickness 0.25 μm Hewlett-Packard) and connection to a FID was used. The column temperature was set at 50 °C for 1 m, then 7 °C/m to 250 °C, and finally at 250 °C for 5 m. The temperature at injection was 240 °C and 250 °C of the detector (split ratio: 1/60). The helium (99.995% purity) was used as a carrier with a flow rate of 1.2 mL/m. The analyzed clove oil volume was 2 μL. The following conditions were used for MS: Detectors: The MS ion source temperature was 250 °C, ionization voltage was 70 eV, ion source temperature was 150 °C, electron ionization mass spectra were acquired over the mass range from 50 to 550 *m*/*z* and scan speed was 769 u/s. Constituent percentages were calculated by electronic integration of FID peak areas.

### 3.3. Electrospinning Solution

For preparation of electrospinning solutions, a mixture of acetic acid and distilled water was used as a solvent. Chitosan alone cannot be electrospun because of its poor chain entanglement [42]; thus, to facilitate the spinning process, PEO was used as a copolymer. Chitosan 5% and 5% PEO solutions were prepared separately using 50:50 of water and acetic acid and then both the solutions were mixed and stirred together for 2 h until a uniform mixture was obtained. 0.5 and 1% CEO was added to the polymer mixture (1:1) and was again mixed for another 2 h to ensure homogenous mixing of the oil.

### 3.4. Fiber Formation

A LINARI-RT Collector Electrospinning machine (Linari Biomedical, Valpiana, Italy) setup was used in this study, which is made up of essential components needed for the electrospinning setup, i.e., a syringe in which the electrospinning solution is filled, injection pump that creates a pressure at the syringe to pull out the solution from the needle at a predefined rate, high-voltage power supply that generated the potential difference between the collector and the syringe for the formation of NFs, and a surface collector. A 5 ml syringe attached with a needle (0.9 mm diameter) was filled with the electrospinning solution and was fixed on the pump. Aluminum foil was used for collecting prepared NFs. Various distance, voltage, and flow rates were used according to DOE, but conditions resulting in appropriate NFs were selected for further processing. NFs were randomly collected on the aluminum foil and it takes approximately 5–6 h for a suitable mat of NFs.

### 3.5. Solution and Process Variables

Solution parameters are very important, as they affect not only the electrospinning but also the formation of fine NFs. Thus, viscosity and conductivity of the blend of the polymers to be spun was measured both before the addition of oil and after the addition of oil at various concentrations [43]. The viscosity was measured with viscometer (Viscotech, VR-3000, Llorenç del Penedès, Spain). Three process parameters, i.e., voltage applied, distance of the needle from the collector, and flow rate of the solution, are important to consider given they largely affect the formation of smooth NFs. The process of electrospinning was carried out at room temperature.

### 3.6. Surface Morphology and Average Fiber Diameter

The morphology of the prepared NFs, both unloaded and loaded with CEO, was analyzed by using FEI-Quanta 250, Czech Republic. The samples of the NFs were attached on an aluminum stub with help of double-sided conducting carbon tape, and morphology was observed after gold-coating with a sputter for 1 min at various accelerated voltages using secondary electron detector mode. On average, 30 NFs were used to calculate the diameter of the fibers by using ImageJ software. The average diameter was calculated and reported with standard deviations. 

### 3.7. Fourier Transform Infra-Red Spectroscopy

Chemical characterization of the prepared NFs mat, CEO, and polymers used was done by Perkin Elmer Spectrum-Two, USA. The prepared NFs mat was first dried to remove any moisture by placing it in a desiccator, then pressed into KBr dishes and analyzed using the ZnSe-based ATR module to confirm the successful incorporation of the oil. The spectra were recorded within the wavelength 600–4000 cm^−1^ using a spectral resolution of 4 cm^−1^ [44].

### 3.8. X-ray Diffraction

Changes in crystal size and the relative crystallinity of NF mat before and after loading with the CEO was done using Malvern PANalyticalX’pert Pro (Netherland) using Cu as X-ray source. Analysis was done at a scanning speed of 1°/min from 10° to 70° (2θ) with incident beam path of 240 mm.

### 3.9. Degree of Swelling and Weight Loss

The capacity of the NFs mat to absorb water is important in the case of wound dressing because it comes in direct contact with the wound and must absorb the exudates from the wounded site. The method already reported [37] was adopted in which a sample was cut according to the required weight and was placed in PBS pH 5.5 for 24 h at 37 °C. pH 5.5 was used to optimize the pH condition of the skin wound. After that, samples were removed and were pressed slightly to remove excess water and were weighed again, and the difference in the weight was noted. DS (%) and weight loss was calculated by using the formula given below:(1)$$DS\;\left(\%\right)=\frac{W_{t}-W_{d}}{W_{d}}\times 100$$
(2)$$Weight\;loss=W_{0}-W_{d}\times 100$$
where *DS* is the degree of swelling, *W_t_* and *W_d_* are wet and dry weights of the mat after submersion in the buffer solution for 24 h, respectively. *W*_0_ is the initial weight of the sample in its dry state. The reported values are the average of three readings.

### 3.10. Percentage Oil Content, Drug-Loading Efficiency (DLE%), and Percentage Yield

10 mg of loaded CEO NFs were dissolved in 50 mL of saline buffer (pH 5.5) with help of magnetic stirring until a clear solution was obtained which was then filtered and diluted adequately [45,46]. The concentration of CEO in the diluted sample was observed using a UV-visible spectrophotometer (UV-1900, BMS, Montreal, QC, Canada) at 279. The content of CEO in NFs was assessed by Equation (3) and the percentage of CEO-loaded in NFs was calculated by using Equation (4).
(3)$$MC=\frac{MA}{MT}\times 100$$
(4)$$ML=\frac{MW}{MN}\times 100$$
where *MC* is the percentage content of CEO, *MA* is the actual amount of CEO, and *MT* is theoretical amount of CEO calculated from drug/excipients ratio. *ML* is CEO percentage in NFs, *MW* is weight of CEO in grams present in NFs, and *MN* is weight in grams of CEO-loaded NFs.

### 3.11. Drug-Release Studies

The in vitro drug-release studies of prepared CS-PEO NFs loaded with oil were performed using cellulose acetate dialysis membrane. Studies were performed using PBS of physiological conditions (pH 7.4) and skin conditions (pH 5.5) [47,48]. NFs mat of known weight were placed in the dialysis membrane, which was clamped on both sides and was immersed in respective PBS at room temperature, stirred at 50 rpm, and further covered with parafilms to avoid any evaporation. After fixed intervals, a small volume of the samples was removed and were immediately replaced with the same amount of fresh PBS. Release studies continued for 48 hours. The samples collected were analyzed for the concentration of oil release using a UV spectrophotometer at 279 nm. The readings were compared with the standard curve. All the readings were taken in triplicate and average readings were reported.

### 3.12. In Vitro Cytotoxicity 

In vitro cytotoxicity was measured using MTT assay. The human fibroblast cell lines were cultured on DMEM medium, which was supplemented with 10% FBS, 1% nonessential amino acids, 2 mM l-glutamine, and 0.1% penicillin-streptomycin at 37 °C in humidified 5% CO_2_ atmosphere. The cells were seed in a 96-well plate at a density of 10,000 cells/well for 24 h before treatment with sample. The prepared nanowebs were sterilized using UV radiation for 45 min and then were immersed in a serum-free medium which was composed of DMEM, 1% lactalbumin, 1% l-glutamine, and 1% antibiotic and antimycotic formulation), and incubated for 24 h to produce extraction media with different concentrations. The extraction media at various concentration was then replaced, and the cells were then re-incubated for 2 and 24 h. After exposure time, the tested extraction solutions were removed, and the cytotoxicity was determined by MTT assay [49,50]. The relative viability (%) was calculated using absorbance at 550 nm determined using a microplate reader (Universal Microplate Analyzer, Model AOPUS01, and AI53601, Packard BioScience, Meriden, CT, USA). The viability of control cells was defined as 100%.

### 3.13. Antibacterial Activity

Antibacterial activity of prepared blank and CEO-loaded NFs was evaluated using the diffusion method. Muller Hinton agar plates were used for the assay. Freshly grown overnight cultures of selected strains (*S. aureus* (ATCC 29213), *E. coli* (ATCC 8739), and *P. aeruginosa* (ATCC 9027)) were prepared in nutrient broth and were spread on the plates with help of a sterilized glass spreader under laminar air flow hood. Appropriate swatches of fibers were prepared and were aseptically placed on MHA plates. Plates were then incubated at 37 °C for 24 h. After 24 h, the zone of inhibition was measured.

### 3.14. In Vivo Wound-Healing

Sprague–Dawley rats were used for in vivo wound-healing studies. Rats with average weight of 250 + 20 gm and average age of 6–7 months were used, were caged separately under a controlled environment (25 + 2 °C, relative humidity 44–56%) with free access to food and water [51]. All the animal studies were carried out after ethical approval from the committee. Rats were shaved and trimmed on the dorsum surface before creating wounds. Ketamine hydrochloride (50 mg/kg, i.p., body weight) was used for anesthesia after which full thickness circular wounds of 1cm x 1cm were created. Studies were carried for a period of 10 days. Animals were divided into 4 groups. Group B was treated with blank NFs. In Group C, CEO NFs were applied. Group D was treated with commercial formulation, while Group A was kept untreated. Average wound contraction was calculated after measuring the wound diameter on the 1st, 5th, and 10th day.
(5)$$\%\;reduction\;in\;wound\;size=\frac{W_{0}-W_{t}}{W_{t}}\times 100$$
where *W*_0_ is the initial wound size and *W_t_* is the wound size after time “t”.

## 4. Conclusions

Clove oil-encapsulated NFs for the topical treatment of wounds were successfully prepared using CS and PEO as polymers. Prepared NFs were characterized using SEM, FTIR, and XRD techniques. The developed NFs showed high encapsulation and high loading of CEO with good release at both pH 5.5 and 7.4 in a biphasic manner with burst release initially followed by sustained release. The NFs show no cytotoxicity against fibroblast cell lines and showed effective antibacterial and wound-healing activity. Thus, the CEO-encapsulated CS-PEO NFs mat can be effectively used as a potential wound-healing candidate.

## Figures and Tables

**Figure 1 molecules-26-02491-f001:**
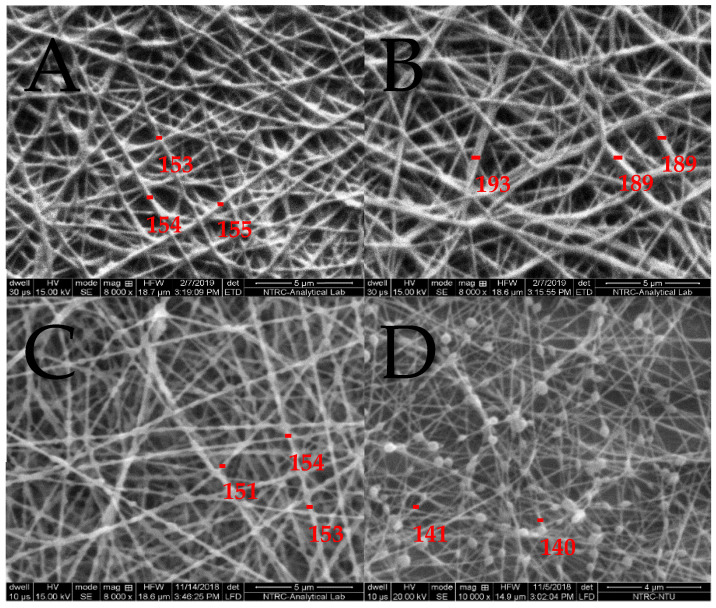
**NFs** formed with 0.5% of CEO (**A**), fibers formed with 1% of CEO (**B**), fibers formed with 60:40 ratio (**C**) and fibers formed with 70:30 ratio (**D**).

**Figure 2 molecules-26-02491-f002:**
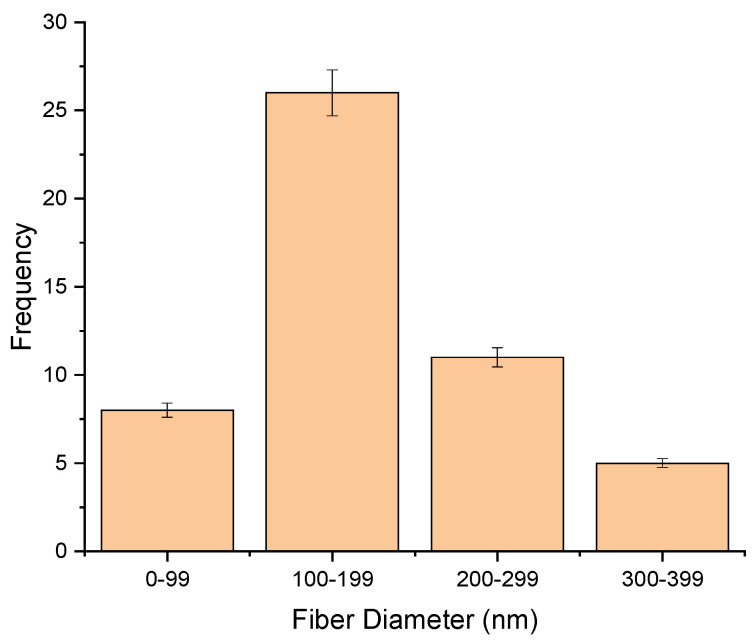
Average fiber diameters measured during formulation.

**Figure 3 molecules-26-02491-f003:**
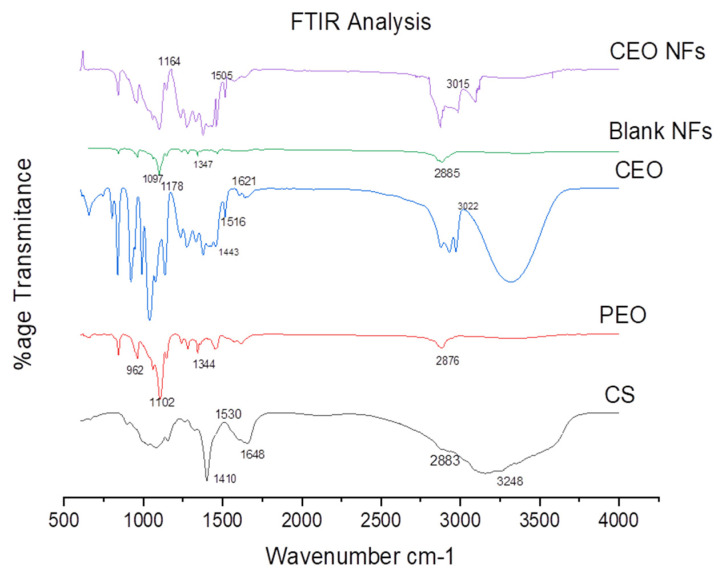
The FTIR spectra of CS, PEO, CEO, Blank NFs, and CEO-encapsulated NFs.

**Figure 4 molecules-26-02491-f004:**
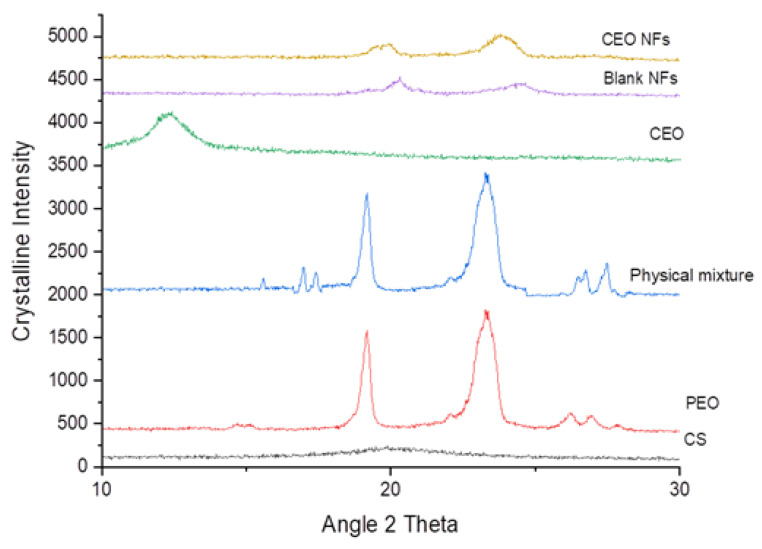
The XRD analysis of CS, PEO, physical mixture (CS-PEO with CEO), CEO, Blank NFs, and CEO-encapsulated NFs.

**Figure 5 molecules-26-02491-f005:**
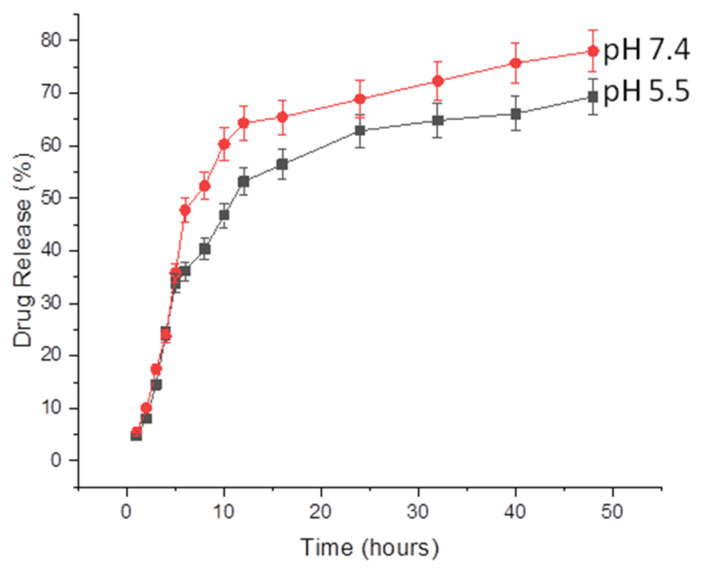
Release of oil from CEO-encapsulated NFs at pH 5.5 and pH 7.4.

**Figure 6 molecules-26-02491-f006:**
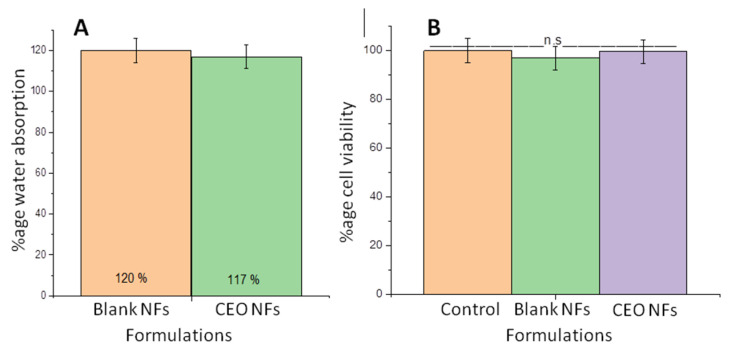
Percentage water absorption by blank NFs and CEO-encapsulated NFs (**A**) and cell viability percentage blank NFs and CEO NFs against fibroblasts cell lines after 48 hours incubation (*n* = 6, (ns = *p* > 0.05) (**B**).

**Figure 7 molecules-26-02491-f007:**
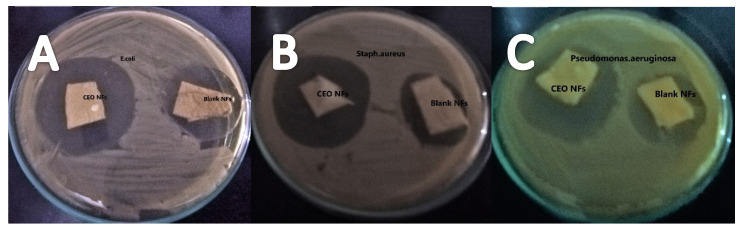
Zone of inhibition of CEO NFs and blank NFs observed against *Escherichia coli* (**A**), *Staphylococcus aureus* (**B**), and *Pseudomonas aeruginosa* (**C**).

**Figure 8 molecules-26-02491-f008:**
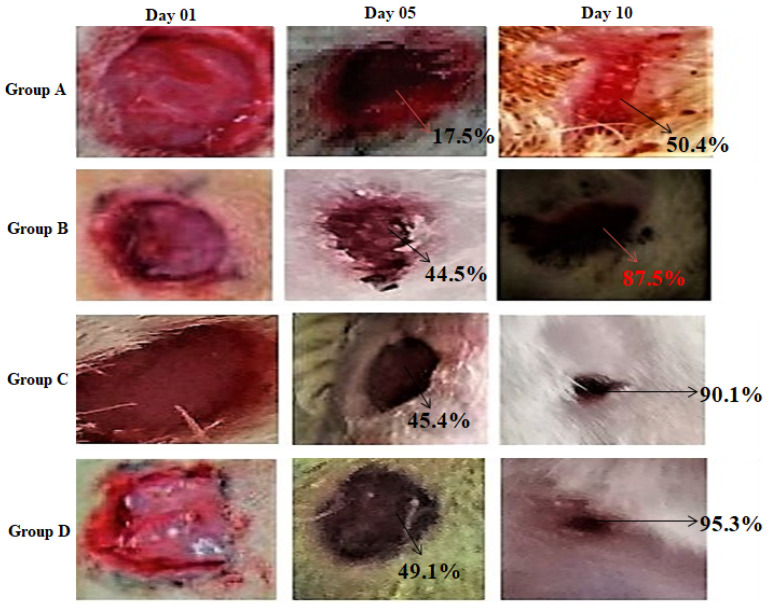
Representative images of wound-healing on various days.

**Figure 9 molecules-26-02491-f009:**
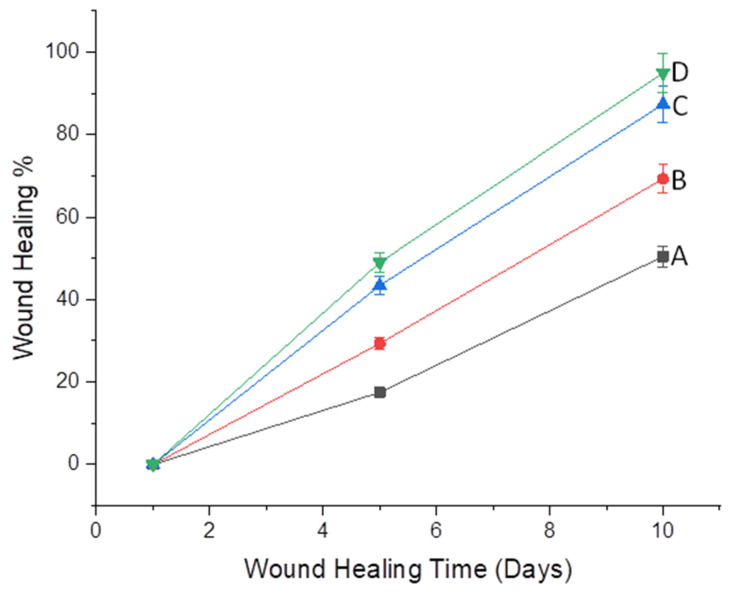
Percentage wound-healing of untreated group (**A**), treated with blank (**B**), treated with CEO-encapsulated NFs (**C**) and treated with commercial product (**D**).

**Table 1 molecules-26-02491-t001:** Percentage composition of various constituents identified in CEO by GC-MS.

S. No.	Constituents	Retention Time (min)	Percentage Composition	S. No.	Constituents	Retention Time (min)	Percentage Composition	S. No.	Constituents	Retention Time (min)	Percentage Composition
1	2-Heptanol acetate	9.28	0.016	12	β-Gurjunene	19.2	0.32	23	α-Farnese	21.32	0.057
2	2-Nonanone	10.23	0.012	13	α-Ylangene	19.42	0.015	24	Eugenyl acetate	21.47	8.32
3	Acetic acid, phenylmethyl ester	11.2	0.06					25	Cadina-1,4-diene	21.53	0.16
4	Methyl salicylate	14.19	0.055	14	α-Humulene	19.81	1.37	26	α-Calacorene	21.77	0.037
5	Chavicol	14.52	0.175	15	Alloaromadendrene	20.13	0.034	27	1-Vinyl-2,6,6-Trimethylcyclohex-1-		
6	α-Cubebene	15.35	0.034	16	δ-Cadinene	20.43	0.26		Ene	21.83	0.57
7	Eugenol	16.48	71.43	17	(*E*)-5-Acetyl-2,2-dimethyl-1-(3’-methyl-1’,3’-yl)bicyclo[2.1.0]pentan	20.67	0.059	28	Caryophyllenyl alcohol	22.04	0.12
8	α-Copaene	17.58	0.53	18	γ-Muurolene	20.89	0.073	29	(−)-Caryophyllene oxide	22.15	0.43
9	Cis-isoeugenol	18.13	0.018	19	α-Amorphene	20.94	0.037	30	Caryophylla-4(12),8(13)-dien-5.beta.ol	22.28	0.027
10	β-Elemene	18.32	0.043	20	α-Muurolene	21.06	0.034	31	Vulgarol B 43 2,3,4-	22.42	0.85
				21	β-Selinene	21.18	0.056	32	Trimethoxyacetophenone	23.13	0.46
11	Caryophyllene	18.86	10.32	22	β –Cadinene	21.27	0.063	33	Benzyl benzoate	24.75	0.49

**Table 2 molecules-26-02491-t002:** Viscosity and conductivity of electrospinning solution at various polymer ratio and oil concentrations.

Sample	Polymer Ratio CS:PEO	Concentration of Oil%	Viscosity	Conductivity	Fiber Morphology
01 02 03 04 05 06	50:50 50:50 60:40 60:40 70:30 70:30	0.5 1 0.5 1 0.5 1	1852 1871 1878 1883 1894 1912	2247 2228 2221 2215 2217 2206	Smooth beadles Smooth beadles Smooth with some beads Beaded Beaded Beaded

**Table 3 molecules-26-02491-t003:** Percentage CEO content, loading, and total yield of CEO NFs.

Formulation Code	CEO NFs
CEO Content (%)	87.6 ± 13.1
CEO loading (%)	8.9 ± 0.98
Total yield (%)	79 ± 9.35

**Table 4 molecules-26-02491-t004:** Percentage wound contraction of wounded skin treated with NFs on different days.

Days	Group A (Untreated)	Group B (Treated with Blank)	Group C (Treated with CEONFs)	Group D (Treated with Commercial Product)
01	0 ± 0 ^ns^	0 ± 0 ^ns^	0 ± 0 ^ns^	0 ± 0 ^ns^
05	17.56 ± 3.4	29.37 ± 2.18 ^ns^	45.41 ± 3.5 **	49.1 ± 3.4 **
10	50.44 ± 4.12	69. 3 ± 2.5 *	90.13 ± 1.5 **	95.3 ± 3.2 ***

Values are given as a mean ± SEM (*n* = 4). Values are compared with the A6 group (untreated) considered to be control. Values are considered to be non-significant (ns = *p* > 0.05), significant (* = *p* < 0.05), more significant (** = *p* < 0.01), and highly significant (*** = *p* < 0.001).

## Data Availability

Data is contained within the article or supplementary material.

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
