# Peer review of "Formulation and Evaluation of a Clove Oil-Encapsulated Nanofiber Formulation for Effective Wound-Healing"

_molecules, 2021, doi:10.3390/molecules26092491_

Round 1

Reviewer 1 Report

  1. The title of the article mentioned “new regenerative sustainable and anti-microbial nanofiber”, but did not specify what kind of raw materials. Please consider whether it is more appropriate to write “Clove oil encapsulated nanofibers” or “CS-PEO NFs” in the conclusion.
  2. In 2.1, please check your expression “The results are given in the Table 1 showing the %age composition and retention time”.
  3. How to use GC-MS to quantitative constituents of clove essential oil? Internal standard or external standard method?
  4. In 2.3, the full English name of “NFs” should be given when it first appear in the article.
  5. In 2.3, the spectrum of clove oil loaded NFs showed characteristic peaks of 3015 cm-1, 1505 cm-1 and 1164 cm-1 in the figure 3. Why the article is written in 3015 cm-1, 1506 cm-1 and 1164 cm-1?
  6. In figure 4, what is physical mixture? It means CS-PEO?
  7. In the subheading 2.5, please make sure "%. age" or "%age", the whole article should be consistent.
  8. I can't understand the Table 3. “%age CEO content, loading and yield” means “% age oil content, Drug Loading Efficiency (DLE %) and %age yield” in the subheading 2.5. The content of the table should correspond to the data. Please redraw and organize Table 3.
  9. Two 2.7 subheading appear in the article. “Water Absorption Capacity” and “Antibacterial Activity”. Please correct it.
  10. “Antibacterial activity” is very important in this article. Where the figure and table can show the “Antibacterial activity”? The content of “Antibacterial activity” is few.

Author Response

  1. The title of the article mentioned “new regenerative sustainable and anti-microbial nanofiber”, but did not specify what kind of raw materials. Please consider whether it is more appropriate to write “Clove oil encapsulated nanofibers” or “CS-PEO NFs” in the conclusion.

Answer: The title of the manuscript is revised as per the suggestions of reviewer.

  1. In 2.1, please check your expression “The results are given in the Table 1 showing the %age composition and retention time”.

Answer: The expressions are corrected.

  1. How to use GC-MS to quantitative constituents of clove essential oil? Internal standard or external standard method?

Answer: Internal standard method was used.

  1. In 2.3, the full English name of “NFs” should be given when it first appear in the article.

Answer: The full name of NFs is provided in its first appearance in the revised manuscript.

  1. In 2.3, the spectrum of clove oil loaded NFs showed characteristic peaks of 3015 cm-1, 1505 cm-1 and 1164 cm-1 in the figure 3. Why the article is written in 3015 cm-1, 1506 cm-1 and 1164 cm-1?

Answer: The peaks of spectrum of clove oil loaded NFs is corrected in the revised manuscript.

  1. In figure 4, what is physical mixture? It means CS-PEO?

Answer: The physical mixture is CS-PEO with CEO.

  1. In the subheading 2.5, please make sure "%. age" or "%age", the whole article should be consistent.

Answer: Corrected throughout the manuscript.

  1. I can't understand the Table 3. “%age CEO content, loading and yield” means “% age oil content, Drug Loading Efficiency (DLE %) and %age yield” in the subheading 2.5. The content of the table should correspond to the data. Please redraw and organize Table 3.

Answer: The table 3 is redrawn and organized.

Table 3. %age CEO content, loading and total yield of CEO NFs

Formulation code

CEO NFs

CEO Content (%)

87.6±13.1

CEO loading (%)

8.9±0.98

Total yield (%)

79±9.35

  1. Two 2.7 subheading appear in the article. “Water Absorption Capacity” and “Antibacterial Activity”. Please correct it.

Answer: The level of heading is corrected in the revised manuscript.

  1. “Antibacterial activity” is very important in this article. Where the figure and table can show the “Antibacterial activity”? The content of “Antibacterial activity” is few.

Answer: The figure of zone of inhibition (antibacterial activity) is provided in the revised manuscript.

Reviewer 2 Report

The paper Formulation and evaluation of a new regenerative sustainable and anti-microbial nanofiber formulation for effective skin wound healing, present novel and interesting results and deserve to be published after several minor improvements:

  1. Please provide high resolution of figure 1. Please measure several fibers on SEM images.
  2. Figure 2 must have error bars, so please add them
  3. The XRD can be smoothed a little bit
  4. Figure 7, please fit it in the page, and provide higher resolution images. Also, add several arrows on this figure for a better focus of attention.
  5. References must be formated according to MDPI style.
  6. Also, please add at least 10 references from 2019-2020 in order to highlight better the importance and the novelty of your work.

After proper improvements, the paper deserve to be published.

Author Response

The paper Formulation and evaluation of a new regenerative sustainable and anti-microbial nanofiber formulation for effective skin wound healing, present novel and interesting results and deserve to be published after several minor improvements:

  1. Please provide high resolution of figure 1. Please measure several fibers on SEM images.

Answer: The figure is revised and fibers are marked

  1. Figure 2 must have error bars, so please add them

Answer: The Figure 2 is revised and has error bars.

  1. The XRD can be smoothed a little bit.

Answer: The figure of XRD is corrected.

  1. Figure 7, please fit it in the page, and provide higher resolution images. Also, add several arrows on this figure for a better focus of attention.

Answer: The figure is revised.

  1. References must be formated according to MDPI style.

Answer: The references are corrected according to the style of MDPI.

  1. Also, please add at least 10 references from 2019-2020 in order to highlight better the importance and the novelty of your work.

Answer: The latest references are incorporated in the revised manuscript.

After proper improvements, the paper deserves to be published.

Round 2

Reviewer 1 Report

OK